# Treadmill Exercise Alleviates Cognition Disorder by Activating the FNDC5: Dual Role of Integrin αV/β5 in Parkinson’s Disease

**DOI:** 10.3390/ijms24097830

**Published:** 2023-04-25

**Authors:** Chuanxi Tang, Mengting Liu, Zihang Zhou, Hao Li, Chenglin Yang, Li Yang, Jie Xiang

**Affiliations:** 1Xuzhou Key Laboratory of Neurobiology, Department of Neurobiology, Xuzhou Medical University, Xuzhou 221004, China; 2Department of Rehabilitation, The Affiliated Hospital of Xuzhou Medical University, 99 West Huaihai Road, Xuzhou 221002, China; 3The College of Medical Technology, Xuzhou Medical University, Xuzhou 221004, China

**Keywords:** treadmill exercise, PD-CI, FNDC5, BDNF, hippocampus, SNpc, synaptic plasticity, DA neuron projection, integrin, CD90

## Abstract

Parkinson’s disease with cognitive impairment (PD-CI) results in several clinical outcomes for which specific treatment is lacking. Although the pathogenesis of PD-CI has not yet been fully elucidated, it is related to neuronal plasticity decline in the hippocampus region. The dopaminergic projections from the substantia nigra to the hippocampus are critical in regulating hippocampal plasticity. Recently, aerobic exercise has been recognized as an effective therapeutic strategy for enhancing plasticity through the secretion of various muscle factors. The exact role of FNDC5—an upregulated, newly identified myokine produced after exercise—in mediating hippocampal plasticity and regional dopaminergic projections in PD-CI remains unclear. In this study, the effect of treadmill exercise on hippocampal synaptic plasticity was evaluated in 1-methyl-4-phenyl-1,2,3,6-tetrahydropyridine (MPTP)-induced chronic PD models. The results showed that treadmill exercise substantially alleviated the motor dysfunction, cognition disorder, and dopaminergic neuron degeneration induced by MPTP. Here, we discovered that the quadriceps, serum, and brain FNDC5 levels were lower in PD mice and that intervention with treadmill exercise restored FNDC5 levels. Moreover, treadmill exercise enhanced the synaptic plasticity of hippocampal pyramidal neurons via increased dopamine levels and BDNF in the PD mice. The direct protective effect of FNDC5 is achieved by promoting the secretion of BDNF in the hippocampal neurons via binding the integrin αVβ5 receptor, thereby improving synaptic plasticity. Regarding the indirect protection effect, FNDC5 promotes the dopaminergic connection from the substantia nigra to the hippocampus by mediating the interaction between the integrin αVβ5 of the hippocampal neurons and the CD90 molecules on the membrane of dopaminergic terminals. Our findings demonstrated that treadmill exercise could effectively alleviate cognitive disorders via the activation of the FNDC5–BDNF pathway and enhance the dopaminergic synaptic connection from SNpc to the hippocampus in the MPTP-induced chronic PD model.

## 1. Introduction

Parkinson’s disease (PD) is one of the most common neurodegenerative diseases globally. In addition to the typical motor symptoms, the clinical manifestations include other non-motor dysfunction: cognitive impairment, sleep disturbance, and gastrointestinal dysfunction [1,2,3]. Parkinson’s disease with cognitive impairment (PD-CI) is the most prevalent, occurring before the onset of motor symptoms and seriously affecting the daily activities and prognosis of people with PD [4,5]. Therefore, in-depth explorations of the pathogenesis and effective methods for treating PD-CI are urgently needed.

Aerobic exercise benefits cognitive function and overall brain plasticity. Moreover, exercise can improve outcomes in neurodegenerative diseases such as Parkinson’s disease (PD) and Alzheimer’s disease (AD) [6,7]. The effects of exercise on the brain are most pronounced in the hippocampus and dentate gyrus, which are important brain regions involved in learning and memory [8]. The benefits of exercise include increased hippocampal size and blood flow, morphological changes in the dendrites and dendritic spines, and increased synaptic plasticity and the induction of myokines [9,10]. As such, exploring the cellular and molecular mechanisms of cognitive functions is considered an important area. Exercise-induced myokine Fibronectin-Type-III-domain-containing 5 (FNDC5) has recently received extensive attention as a modulator of exercise’s beneficial effects on the brain [11]. FNDC5 is secreted and isolated from muscles during exercise, is cleaved as a transmembrane signal peptide to produce the soluble form of Irisin, and regulates metabolism [12]. Wrann et al. observed that exercise regulates the hippocampus’s FNDC5 expression and release of brain-derived neurotrophic factor (BDNF) [13]. Decreases in BDNF expression causes an imbalance in neuronal functioning and survival in neurodegenerative disorders. The above suggests that the protective effect of exercise on cognitive function may be related to the direct upregulation of BDNF in the hippocampus by FNDC5, which may increase synaptic plasticity. Therefore, the question remains regarding the specific molecular mechanism through which FNDC mediates BDNF expression in the hippocampus.

PD is characterized by the progressive loss of dopaminergic (DA) neurons in the substantia nigra pars compacta (SNpc) region. Pathological DA neuron loss reduces dopamine neurons’ projection into the hippocampus, meaning anatomical fiber loss and impairments in aversive memories [14]. The reduction of dopamine-derived projections in the hippocampus may mediate the decline in synaptic plasticity, producing some symptoms of cognitive deficits can be ascertained to a certain extent [15]. Treadmill training (moderate-intensity aerobic exercise) reversed the loss of dopaminergic neurons in mice with Parkinson’s disease [16] through reducing the spread of α-synuclein [17]. Additionally, moderate-intensity aerobic exercise inhibited NLRP3 inflammasome activation to protect neurons [18]. Hence, we also hypothesized that treadmill exercise can actively results in hippocampal neuronal protection by increasing the density or contact point quantity of dopaminergic synaptic knobs in the hippocampus.

In the current study, we followed the frequency, intensity, time, type (FITT) principle for moderate-intensity aerobic exercise (treadmill exercise: 10 m/min, 60 min/day, and 5 days/week for a total of 8 weeks). Next, we aimed to determine the effect of treadmill exercise on FNDC5 levels, an important exercise-related factor in the periphery, and its expression in the central nervous system and the cognitive performance of PD mice. Next, we sought to clarify how FNDC5 mediates enhanced hippocampal plasticity and plays a role in the protection of cognitive function in Parkinson’s disease. 

## 2. Result

### 2.1. Treadmill Exercise Rescues MPTP-Induced Loss of Dopaminergic Neurons

The detailed animal study timeline is shown in Figure 1. By comparing the long-term MPTP administration group with the control group, we observed the number of TH-positive neurons, and the TH expression in the SNpc significantly decreased (Figure 2), which suggested that we successfully constructed an MPTP-induced chronic PD model. Moreover, the results of behavioral experiments also confirmed this success, as shown in Figure 3 and Figure 4. To identify the neuroprotective effects of treadmill exercise, Western blotting and immunofluorescence (IF) assays were used to observe tyrosine hydroxylase (TH) protein levels and the number of TH positive cells in the SNpc (Figure 2A,D). Briefly, the MPTP mice showed decreased TH expression (*p* < 0.001) and reduced TH-positive neurons (*p* < 0.001) compared with those in the control group (Figure 2B,E). Those in the MPTP+EX group were higher than in the MPTP group (*p* = 0.029, *p* < 0.001) (Figure 2B,E). Moreover, ELISA was performed to detect the expression changes of dopamine neurons in the SNpc, which presented an outcome (*p* < 0.001) similar to that shown via IF (Figure 2C). These findings suggested that treadmill exercise rescued the loss of DA neurons in the MPTP-induced mice.

### 2.2. Treadmill Exercise Protects against Motor Deficits in MPTP-Induced Chronic PD Model

The open-field and pole tests were used to determine the locomotor activity and equilibrium coordination of the mice, respectively. In the open-field test, the MPTP mice that performed a treadmill intervention demonstrated a significant improvement in the total travelled distance (*p* < 0.001) and entries in the center zone (*p* < 0.001) compared with the mice of the MPTP group (Figure 3A–C). The results of the pole test revealed that the T-turn (*p* < 0.001) and T-total (*p* < 0.001) time were significantly longer in the MPTP group than in the control group. Instead, those above (*p* < 0.001) (*p* < 0.001) were shorter in the MPTP+EX group (Figure 3D,E). These results revealed that treadmill exercise could ameliorate MPTP-induced motor deficits.

### 2.3. Treadmill Exercise Protects against MPTP-Induced Cognition Dysfunction

The changes in the spatial learning and memory of the mice in each group were detected with the Morris water maze (Figure 4). Figure 4A shows representative trajectories from every group of mice. During the learning phase, the path length and time spent finding the platform in all groups gradually reduced (Figure 4D,E). After the platform was removed, we observed a decrease in the total travel distance to the target quadrant and the times crossing the platform in the MPTP group compared with the control group (*p* < 0.001). All decreases were reversed in mice subjected to treadmill exercise (*p* < 0.001) (Figure 4B,C). These results indicated treadmill exercise improved spatial learning and memory deficits in MPTP-induced chronic PD model. A novel object recognition test was conducted to compare the differences in memory ability and exploration ability of each group of mice. (Figure 4F). The outcomes showed significant restoration both in total exploration time (*p* = 0.001) and exploration time with a novel object (*p* < 0.001) in the mice of the MPTP+EX group compared with the mice in the MPTP groups (Figure 4G,H). Thus, we concluded that treadmill exercise has a positive influence on the anti-MPTP-induced exploration and memory impairment effects.

### 2.4. Effects of Treadmill Exercise on Serum and Brain FNDC5 Levels in MPTP-Induced Chronic PD Model 

As treadmill exercise increased the secretion of myokines, we then examined the content of FNDC5 in the serum, muscle, and brain using ELISA and Western blotting tests. The MPTP mice receiving the treadmill intervention showed an increase in FNDC5 protein levels in the whole brain and muscle compared with those in the MPTP group (Figure 5A,B). Similar outcomes were observed after serum FNDC5 was detected with ELISA (Figure 5E). These findings clarified that general changes in the FNDC5 levels occurred in mice’s muscle, serum, and brain after the treadmill exercise intervention. To further determine which brain region was closely associated with the FNDC5 level in this study, we detected the content of FNDC5 in multiple regions, including the prefrontal cortex (PFC), substantia nigra pars compacta (SNpc), striatum (STR), hippocampus (Hi), and cerebellum (CERE). We found that the contents of FNDC5 in the substantia nigra and hippocampus were higher than in the other regions (Figure 5C,D). Moreover, correlation analysis was performed to associate the serum FNDC5 content with the cognitive function-related indices of the previous behavior tests. We found a positive correlation between the FNDC5 content and exploration time with the novel object (R^2^ = 0.36, *p* < 0.001) and time spent in the target quadrant (R^2^ = 0.19, *p* = 0.007) (Figure 5F). Taken together, these results demonstrated that the content of FNDC5 correlated with the cognitive function of the mice.

### 2.5. Treadmill Exercise Enhanced Synaptic Plasticity of Hippocampal Pyramidal Neurons via Increased Dopamine Levels and BDNF in PD Mice

As treadmill exercise may play a vital role in synaptic plasticity, we further explored the possible mechanism through which treadmill exercise affected MPTP-induced mice in this study. Based on previous routine groups, we set two new groups: ME+BDNF inhibitor (ME-CTX-B) and ME+DA blocker (ME-CPZ). We first detected the dopamine level in the hippocampus. We found that the ME-CPZ group had lower dopamine levels than the MPTP+EX group (Figure 6A). The outcome showed that the DA level was downregulated by its inhibitor-CPZ. Next, the expression of synaptic-related proteins was detected with Western blotting. The results showed that the expressions of PSD-95, synapsin, SNAP47, synaptophysin, TH, and BDNF in the hippocampus of the MPTP group were lower than in the control group (Figure 6B–H). Instead, the expressions of the above proteins in the MPTP+EX group was higher than those in the MPTP group. In addition, the expressions of synaptic-related proteins in the MPTP+EX group with BDNF inhibitor and DA blocker were lower than in the ME group. These results suggested that treadmill exercise may increase the synaptic plasticity of hippocampal pyramidal neurons via upregulating the DA and BDNF levels in MPTP-induced mice. 

To verify the effect of treadmill exercise on the synaptic plasticity of the MPTP-induced mice, we detected the synaptic density and ultrastructure of the hippocampal CA1 region via transmission electron microscopy (TEM). Compared with the MPTP group, the synaptic active region’s length and the postsynaptic dense material’s thickness were significantly increased in the MPTP+EX group, and width of the synaptic cleft was shorter (Figure 6I–L). Subsequently, the morphology of pyramidal neurons in the hippocampal CA1 region was observed by Golgi staining. In the control group, the vertebral cells located in the shallow layer produced many processes to form the basal dendrites. The branches of the basal dendrites spread to the radiation layer and extended horizontally, presenting a close arrangement of several layers. Here, we found that a few dendrites had only one or two branches, and dendritic spines fell off or even disappeared on the branches in the MPTP mice. After treadmill exercise, the number of dendrites and axons was significantly higher than that in the MPTP group, and the dendrite branches and complexity significantly increased compared with or were even comparable to those in the control group (Figure 6M,N). 

### 2.6. Exercise-Induced FNDC5 Directly Activated Integrin-Mediated Upregulation of BDNF in Hippocampus 

As an exercise-induced key factor, FNDC5 was closely related to the cognitive function of the mice in the current study. To further explore its specific molecular mechanisms of this effect, the mice in the MPTP+EX group were given an additional FNDC5 administration. Subsequently, a Western blot assay was used to detect the protein expression of the FNDC5–integrin–BDNF pathway. We found that the FNDC5, integrin αV, integrin β5, and BDNF protein expressions in the hippocampus after FNDC5 administration were significantly higher compared with those in the M and ME groups (*p* < 0.05) (Figure 7A–E). Next, a coimmunoprecipitation (Co-IP) assay was conducted to verify the interaction between FNDC5 and integrinαVβ5. After the FNDC5 intervention, the binding ability of the two proteins was significantly enhanced compared with that of the other two groups (*p* < 0.05) (Figure 7H,I). Then, HT22 hippocampal neurons were cultured and treated with FNDC5, and we used immunofluorescence to observe the changes in the integrin receptor (Figure 7F–G). The results revealed that the integrin fluorescence intensity and distribution in the hippocampal neurons after FNDC5 intervention were stronger and higher, respectively, than those in the blank control group (*p* < 0.05) (Figure 7F,G). 

### 2.7. FNDC5 Restored Projection of Substantia Nigra to Hippocampus, Promoting Dopaminergic Synaptic Connection in the Hippocampus via Interaction between Integrin and Thy-1/CD90

FNDC5 can directly activate the integrin-mediated upregulation of BDNF in the hippocampus. Additionally, as PD is characterized by a massive loss of dopamine neurons, a stable amount of DA neurons is crucial to slowing the course of PD. Figure 6 shows that treadmill exercise increased the synaptic plasticity of the hippocampal pyramidal neurons by upregulating the DA and BDNF levels in the MPTP-induced mice. We wondered whether FNDC5 mediates an indirect pathway to promote an increased projection of dopamine neurons in the SNpc into the hippocampus. Thus, we verified this by injecting the anterograde virus into the SNpc region (Figure 8A,B). We subsequently captured stronger projection signaling in the hippocampus after FNDC5 treatment than in the other two groups (*p* < 0.05) (Figure 8C,D). With confocal microscopy, we determined the branches of dopamine neurons in different groups. We found that the neurons in the mice with FNDC5 administration had more branches and sprouting dots compared to the ME groups (*p* < 0.05) (Figure 8E,F).

To further investigate the mechanism through which FNDC5 acts on dopamine neurons, we first demonstrated that exercised-induced FNDC5 promoted the survival of dopaminergic neurons in the substantia nigra. As shown in Figure 8C–F shown, we found that after exercise, especially after exogenous additional supplementation with FNDC5, the numbers of dopaminergic neurons and dopaminergic synaptic connections projecting to the hippocampus significantly increased (Figure 8C,E, *p* < 0.001). We extracted tissue protein from the hippocampus to explore the molecular mechanisms of the dopaminergic synaptic connections regulated by exercise. Thy1/CD90 is highly expressed on the surface of adult neurons and is thought to play a key role in regulating neurite extension and synaptic connectivity. We therefore examined the binding ability between integrin αVβ5 and Thy1/CD90 in the hippocampus though immunoprecipitation. The results indicated that the binding ability was significantly enhanced in the ME group treated with FNDC5 compared with that in the ME mice (*p* < 0.01) (Figure 8E,F).

## 3. Discussion

In the current study, we found that treadmill exercise rescued the massive loss of dopaminergic neurons and restored cognition deficits in MPTP-treated mice. The potential mechanism of this effect is that treadmill exercise increases synaptic plasticity via activating the FNDC5–BDNF pathway. Moreover, FNDC5 effectively promoted DA neuron projection into the hippocampus (Figure 9). Thus, our evidence suggests that treadmill exercise and myokine FNDC5 exhibit promise for practical application to mitigate Parkinson’s disease with cognitive dysfunction (PD-CI) in the future.

As a common PD-related symptom, PD-CI has received attention as it appears earlier than clinical indications and seriously affects patients’ daily life and prognosis [4]. Aerobic exercise benefits the whole body and improves the prognosis of neurodegenerative diseases, especially overall cognitive status [19]. In terms of the clearance of pathological proteins, moderate-intensity exercise (5–12 m/min bursts for 30 min executed 6 days/week) can significantly inhibit α-syn from spreading to the SNpc and cortex of PFF (α-syn preformed fibrils)-seeded mice brain to relieve the effects of PD. Since its discovery, researchers have been continuously exploring the role of the hippocampus in cognitive function. This is essential for understanding the cellular and molecular mechanisms of cognitive function and the specific region of the brain involved in learning and memory. In this study, we explored the mechanisms underlying the effects of treadmill exercise on the cognition of PD-induced mice by establishing a chronic MPTP mouse model. In MPTP-induced mice, we observed that the progressive loss and degeneration of dopaminergic neurons in the SNpc led to varying degrees of motor dysfunction and cognition deficits. Additionally, our results showed that compared with chronic-MPTP mice, the numbers of TH-positive DA neurons and the TH protein expression in the SNpc significantly increased after receiving treadmill exercise. As mentioned before, similar outcomes were observed in that treadmill training could effectively mitigate the motor deficits and cognitive dysfunction experienced in MPTP mice. 

The exercise-induced benefits were mainly reflected in the increases in hippocampus size and blood flow, morphological changes of the dendrites, increases in synaptic plasticity, decreases in neuroinflammation, and the promotion of exercise-related factors. FNDC5/irisin is a factor secreted by skeletal muscle during exercise. FNDC5 is a transmembrane signal peptide and is cleaved to assume its soluble form as Irisin, producing benefits: regulating metabolism and promoting bone and cardiovascular protection [20,21]. In addition to skeletal muscle, irisin can be secreted in various brain regions, such as Purkinje cells in the cerebellum, astrocytes in the hippocampus, neurons in the brain, or cerebrospinal fluid [22,23]. The contents of FNDC5 in the skeletal muscle, peripheral blood, and brain were also detected in this study. The results of Western blotting and ELISA showed that compared with the MPTP group, the contents of FNDC5 in the skeletal muscle, peripheral blood, and brain after exercise training in the MPTP+EX group were significantly higher.

The knockout of FNDC5 in neuron precursors disrupted the development of mature neurons, which indicates the developmental role of FNDC5 in neurons [24]. Recently, the role of irisin in combating neurological dysfunction has been explored. Irisin enhances neurogenesis, cell proliferation, and synaptic plasticity [25,26]. Here, we speculated that the expression of FNDC5 might be related to cognitive function. By establishing a correlation analysis between the serum FNDC5 content and the cognitive performance index, we finally revealed a positive correlation between the FNDC5 content and cognitive function. 

Evidence shows that FNDC5 regulates the expression and release of brain-derived trophic factor (BDNF) in the hippocampus, affecting cognitive function [13]. In this study, we found that FNDC5 administration increased binding to the receptor integrin, thereby activating BDNF expression. As a member of the neurotrophic factor family, BDNF is distributed in a wide range of regions, such as the central nervous system, peripheral nervous system, endocrine system, bone, and cartilage tissue, but is mainly expressed in the central nervous system, especially in the hippocampus and cortex [27]. BDNF binds to the receptor tyrosine kinase B (TrkB), which activates the RAS–MAPK pathway and finally activates CREB. As such, it plays a critical role in promoting synaptic plasticity, neurogenesis, and cell survival. The promotion of cell survival is mainly reflected in maintaining and promoting the differentiation, growth, and regeneration of various neurons, especially 5-hydroxytryptamine (5-HT) and dopaminergic neurons [28,29]. The above descriptions show that BDNF is a prime therapeutic candidate for various neurological disorders. CTX-B, a highly potent and selective TrkB inhibitor, can effectively reverse the effect of BDNF. BDNF is regarded as an important factor for neurogenesis, synaptic plasticity, and neuronal network organization in brain circuits [30]. BDNF interacts with the TrkB receptor to maintain dendritic spine formation and the morphology of pyramidal neurons [31]. As such, when BDNF is blocked by CTX-B, the synaptic-plasticity-promoting effects of this pathway are inhibited. BDNF can be delivered into the brain as a nanoformulation to exert its neuroprotective effects in clinical transformation application. In our study, according to the results of electron microscopy and Golgi staining, we observed ultrastructural changes in the hippocampal CA1 region. The results indicated that treadmill exercise increased the synaptic active region’s length, postsynaptic density, and the number of spines on the dendritic spines and shortened the synaptic cleft in PD mice. In addition, a significantly increased concentration of synaptic-related protein was observed in the MPTP+EX group. 

DA deficits mainly trigger the clinical brain symptoms of people with PD. The MPTP-induced degeneration of DA neurons resulted in decreased contents of TH, DA, and DOPAC in the substantia nigra and striatum as well as in the hippocampus [14]. Furthermore, consuming DA in the hippocampus may alter synaptic transmission and activity-dependent synaptic plasticity changes. Therefore, we assumed that the massive loss of dopaminergic neurons in the SNpc would further reduce their projection into the hippocampus, which might be one of the mechanisms through which PD-CI occurs. The hippocampus is anatomically controlled by the projections of dopaminergic neurons in the midbrain, and the release of DA in the hippocampus may regulate hippocampal plasticity [32]. In this study, we used a DA antagonist (CPZ) to block exercise-induced hippocampal DA increases. In this group of mice, synaptic plasticity was reduced in the hippocampus, and the expression of synapse-associated proteins was downregulated. Similarly, we previously observed that efficient regional dopamine transmission is crucial for maintaining the plasticity of pyramidal neurons [33]. Dopamine release increase and D2 receptor signaling activation in the hippocampus enhanced episodic-like memory and working memory through the associated changes in hippocampal synaptic strength [34]. Therefore, after treatment with CPZ, which prevents D2 activation, the mice presented low hippocampal plasticity. Moreover, the loss of dopaminergic innervation could lead to abnormalities in hippocampal networks [35]. 

In this study, we observed changes in the DA projection into the hippocampus via injecting lentivirus at the SNpc region, and we found that reduced hippocampal projection could be rescued in the FNDC5-treated group. As adhesion receptors for extracellular ligands, integrins transduce biochemical signals into cells via downstream effector proteins [36]. Integrin may act as a receptor for FNDC5 in bone and fat cells [37]. The subsequent release of Irisin reversed intestinal injury, and receptor integrin expression was detected in intestinal epithelial cells [38]. We also observed integrin upregulation in the hippocampus, suggesting that integrin binds to some cell junction molecules, thereby increasing the connection of dopaminergic neurons to the hippocampal pyramidal neurons. Thy1/CD90, a GPI-anchored development-regulated cell surface antigen protein, interacts with integrin to mediate bidirectional intercellular communication [39]. It is highly expressed on the surface of adult neurons and is thought to regulate adhesion and migration activities such as neurite extension [40]. As we hypothesized, similar results were found: FNDC5 promotes the interaction of the integrin of pyramidal neurons and the CD90 of dopaminergic neurons.

The current study confirmed that treadmill exercise reversed cognition dysfunction by activating the FNDC5–BDNF pathway and up-regulating DA neurons. Furthermore, treadmill exercise could also activate the FNDC5–BDNF pathway directly and enhance synaptic plasticity. Treadmill exercise also increased dopaminergic synaptic connection with the hippocampus via enhancing the interaction of integrin αVβ5 and CD90. The above shows evidence supporting treadmill exercise as a nonpharmacological therapeutic strategy for PD and FNDC5 as a clinical prognostic biomarker.

## 4. Material and Methods

### 4.1. Animals and Experimental Design

All animal experiments were conducted following the guideline of the Care and Use of Laboratory Animals published by the United States National Institutes of Health and the Institutional Animal Care and Use Committee of Experimental Animal Center of Xuzhou Medical University (approval Nos. L20210226140 [2 March 2021] and L2022115018 [28 November 2022]). Male C57BL/6J mice (7 weeks, weighing 22–25 g) were obtained from the Experimental Animal Center of Xuzhou Medical University. All mice had free access to food and water at room temperature (25 ± 2 °C) with 50% relative humidity and under a 12 h light/dark cycle and pathogen-free conditions. After acclimation, the animals were randomly divided into three groups: saline injection control, MPTP injection, and MPTP injection with treadmill exercise (MPTP+EX). MPTP (Cat#M0896, Sigma Aldrich, St. Louis, MO, USA) (25 mg/kg in saline, i.p.) and probenecid (Cat#PHR2608, Sigma Aldrich, St. Louis, MO, USA) (250 mg/kg in dimethyl sulfoxide, i.p.) were injected into the mice twice per week for five weeks to establish a chronic PD mice model [41].

### 4.2. Treadmill Exercise Protocol and Drug Administration

A treadmill (6 lanes, Model JD-PT, Jide Instruments, Shanghai, China) was adopted in this study. The treadmill exercise protocol was performed as previously described [42]. For the exercise program adopted in this study, we referred to the exercise frequency, intensity, time, type (FITT) principle, which is moderate-intensity aerobic exercise: it is a treadmill training program widely used in animal models. Before MPTP treatment, the MPTP+EX group was trained in 8 m/min intervals for 30 min/day for five days to familiarize the mice with the environment. After one week of adaptation, mice continued to exercise (12 m/min, 60 min/day, 5 days/week for 8 weeks). At the end of the third week, these mice were given MPTP treatment (Figure 1). Without running, the mice in the control and MPTP groups remained on the treadmill for 30~60 min. The above drugs were prepared by dissolving them in 0.9% saline; they were administered intraperitoneally.

In the MPTP+EX group, cyclotraxin-B (CTX-B, BDNF antagonist, manufactured as Tat-cyclotron-B; Bio S&T, Montreal, QC, Canada) was administered i.p. with a double-injection protocol (20 mg/kg with a 90-min interval) (n = 12), as designed by Cazorla et al. [43,44]. Chlorpromazine hydrochloride powder (CPZ, C8138-5G, Sigma-Aldrich (Shanghai, China) Trading Co. Ltd., Canada, dopamine receptor antagonists) was dissolved in physiological saline and administered i.p. (2 mg/kg) (n = 12) [45,46]. The drugs were given with a double 90 min interval i.p. injection three hours before being sacrificed. Mice were intravenously administered 250 μg/kg FNDC5 (067-29A; Phoenix Pharmaceuticals, Inc., Burlingame, CA, USA) (n = 6) [38]. The drug was given for 7 days during the last week of the schedule.

Experienced operators performed all treadmill exercises and drug administration to minimize animal suffering. Behavior tests were performed the next day after animals completed the treadmill exercise intervention. For following experiments, animals were anesthetized with 40 mg/kg sodium pentobarbital and killed.

### 4.3. Cell Culture and FNDC5/Irisin Treatment

HT22 cells and MES23.5 were cultured in MEM medium (A1049001; Gibco, New York, NY, USA) supplemented with 10% fetal bovine serum (FBS.10099141; Gibco, New York, NY, USA) and a 100 units/mL penicillin/streptomycin mixture (15 070 063; Gibco, New York, NY, USA). They were administered with 10 nmol/L FNDC5 for 90 min to create the FNDC5 intervention group.

### 4.4. Behavior Tests

#### 4.4.1. Open-Field Test

The test area generally consisted of a white plastic rectangular box (50 cm (L) × 50 cm (W), × 50 cm (H)) and a camera used to monitor mouse movement. After acclimation, each mouse was placed in the center of the open field for 5 min in normal lighting. Then, the box was cleaned with 75% alcohol to avoid odor interference. Parameters observed in the open field test included total distance traveled and entries in the center zone.

#### 4.4.2. Pole Test

The pole test is a simple test used to evaluate the degree of bradykinesia following MPTP treatment. Each mouse was placed on top of a rough-surfaced pole with its head upward (1 cm in diameter and 50 cm in height). The time from head-up to head-down (T-turn) and the time the mouse spent at the bottom of the pole with its posterior limbs touching the ground (T-total) were recorded. The mouse was tested 3 times with a 10 min interval rest. The average value was calculated for statistical analysis.

#### 4.4.3. Morris Water Maze (MWM) Test

The MWM experiments were conducted to examine learning and memory. The Morris water maze testing consisted of a blue circular pool with a diameter of 120 cm and a height of 30 cm, filled with 24 °C water. The maze was divided into four quadrants: north, east, south, and west. The learning phase involved 4 trials per day for 5 consecutive days. For each trial, the mice were allowed to swim for 60 s to find the hidden platform. Then, mice were allowed a 10 s rest period on the platform, assuring that each mice had equal time to observe spatial cues after each trial. In this stage, the latency time and path length were recorded daily. A probe trial was performed the day after the last training day to assess spatial memory retention: the platform was removed from the pool, while all other factors remained the same. During the probe trial, each mouse was placed in an area of the pool from the same quadrant and allowed to swim for the 60 s. The percentage of time spent in the previous target quadrant and the number of crossings over the platform location were recorded.

#### 4.4.4. Novel Object Recognition (NOR)

A NOR test was conducted in three phases to examine drug and treadmill exercise effects on recognition memory through habituation, sample, and test phases, as previously described [47]. For habituation, every mouse was placed in the empty box for 10 min on 2 consecutive days. During the sample phase, two identical objects (objects A and A) were placed in the middle area of the arena, and the mouse was allowed to freely explore the box 3 times for 5 min. Then, a novel object replaced one of the familiar objects in the box (objects A and B). The mouse was allowed to explore the objects for 5 min, and software (ANY-maze version 7.1, Stoelting, Wood Dale, IL, USA) was used to record the sniffing time. The preference score for the novel object was calculated by using the formula: the time spent with the novel object/the total object exploration time × 100%.

### 4.5. Detection Methods

#### 4.5.1. Immunofluorescence Staining

For immunofluorescence staining analysis, extracted whole-brain tissues were post-fixed in 4% PFA for 24 h and then dehydrated with 30% sucrose solution for 48 h. Then, 20 μm brain sections were obtained with a freezing microtome. The sections were rinsed with phosphate-buffered saline (PBS) 3 times for 5 min and blocked with 5% BSA for 2 h at room temperature. Free-floating sections were incubated with the primary antigen of mouse anti-TH (1:1000, ab112, Abcam, Shanghai, China) overnight at 4 °C. Next, the sections were washed 3 times in PBS for 5 min each and incubated with secondary antibody: goat anti-mouse IgG H&L-Alexa Fluor^®^ 488 (1:800, ab150113, Abcam, Shanghai, China). DAPI was used to stain the cell nuclei (Melone, MA0222). 

HT22 cells were similarly fixed in 4% paraformaldehyde, followed by the detection of primary antibody rabbit integrin αVβ 5 (1:200, sc-81632, Santa Cruz, Dallas, TX, USA), secondary antibody donkey anti-rabbit IgG H&L Alexa Fluor^®^ 594 (Thermo Scientific, Waltham, MA, USA, A21207, 1:800), and DAPI staining. Fluorescence images were captured using a fluorescent microscope (Olympus, Tokyo, Japan). The number of positive immunoreactivities cells and fluorescence intensity were analyzed using ImageJ software (National Institute of Health, Bethesda, MD, USA, ImageJ https://imagej.nih.gov/ij/).

#### 4.5.2. Western Blotting

To extract protein samples from tissues, we followed a previously described experiment protocol [48]. The SNpc/midbrain samples extracted from the brains of mice were homogenized in RIPA lysis buffer, and the protein concentration was determined with a bicinchoninic acid protein assay kit (BCA, 23235, Thermo Fisher Scientific, Waltham, MA, USA). The samples were separated with SDS–polyacrylamide gel (10–12%) electrophoresis for 90 min and then transferred to nitrocellulose filter membranes at 80–120 V stored for 1 h. The membranes were washed 3 times in PBS for 5 min each, and then they were blocked with 5% bovine serum albumin (BSA) for 2 h at room temperature. The membranes were incubated with different primary antibodies at 4 °C overnight. These primary antibodies included TH (1:1000, ab112, Abcam, Shanghai, China), PSD95 (1:1000, 3450S, CST, Fall River, MA, USA), Synapsin (1:1000, NB300-104, NOVUS (Shanghai) Co. Ltd, Centennial, CO, USA), CAMKII(1:1000, 50049S, CST, Fall River, MA, USA), SNAP47 (1:1000, ab172609, Abcam, Shanghai, China), Synaptophysin (1:1000, ab52636, Abcam, Shanghai, China), BDNF (1:200, ANT-010, Alomone Labs, Jerusalem, Israel), FNDC5 (1:1000, bs-8486R, Bioss, Beijing, China), integrin αV (1:200, sc-9969, Santa Cruz, Dallas, TX, USA), integrin β5 (1:200, sc-398214, Santa Cruz, Dallas, TX, USA), and β-actin (1:5000, 60008-1-Ig, Proteintech, Wuhan, China). After that, they were washed and incubated in secondary antibody: anti-mouse IRDye^®^ 680RD-conjugated antibody (1:10,000) or anti-rabbit IRDye^®^ 800CW-conjugated antibody (1:10,000) for 2 h. An imaging system (LI-COR Biosciences) was used to visualize protein bands, and the grayscale values were analyzed with Image J software (National Institutes of Health, NIH, Bethesda, MD, USA, ImageJ https://imagej.nih.gov/ij/).

#### 4.5.3. Enzyme-Linked Immunosorbent Assay (ELISA)

Mice were killed after the behavior test, and brain and blood samples were collected. The brain samples were homogenized and then centrifuged with the blood samples at 1500× *g* for 10 min. The supernatant was harvested and stored at −80 °C until analysis could be performed. ELISA kits were used to measure serum FNDC5 (E-EL-M0392c, Elascience) and DA levels in the midbrain and hippocampus (Cat#JL11187-48T), according to the instructions for the kits.

#### 4.5.4. Co-Immunoprecipitation (Co-IP)

Co-IP was conducted as previously described. Briefly, the cell lysate was combined with 1.2mL of dilute buffer (lysis buffer), which was incubated with Protein A/G Agarose (sc-2003; Santa, Dallas, TX, USA) and mouse anti-integrin alpha V/beta 5 antibody (sc-81632; Santa, Dallas, TX, USA) overnight at 4 °C. The following day, the agarose beads were collected via centrifugation at 5000 rpm for 15 min at 4 °C. Next, the beads were eluted two to three times and boiled to denature the protein–bead complex. The proteins were separated via SDS-PAGE, electrophoretically transferred onto 0.22 μm nitrocellulose membranes, and immunoblotted with the primary antibodies (rabbit anti-fndc5 antibody (1:1000, bs-8486R; Bioss, Beijing, China), mouse anti-integrin alpha V antibody (1:200, sc-9969; Santa, Dallas, TX, USA), and mouse anti-integrin beta 5 antibody (1:100, sc-398214; Santa, Dallas, TX, USA)).

#### 4.5.5. Golgi Staining

To visualize the entire architecture of a neuron, we used a Golgi-Cox staining kit following the manufacturer’s protocol (FD Rapid GolgiStain^TM^ Kit, FD NeuroTechnologies, Inc.) with minor alterations. In brief, mice were anesthetized with 40 mg/kg sodium pentobarbital before killing and transcardially perfused with cold 4% saline for 3 min. The animal brains were removed from the skull as quickly as possible. Then, the tissue blocks were immersed in an impregnation solution (mixture of equal volumes of solutions A and B, prepared 24 h before use) at room temperature (RT) for 2 weeks, with the impregnation solution replaced each following day. Next, the blocks were transferred into solution C at RT. The buffer was replaced on the next day, and the sample was stored for 4 days in the dark. The brain was cut into sections (100 μm) on a vibratome and then immediately mounted on gelatin-coated microscope slides by dropping solution C. After naturally air-drying at RT overnight and 5 min washes with Milli-Q water, the sections were incubated in the working solution (a ratio 1:1:2 of solution D: E: double-distilled water) with gentle shaking for 10min. After dehydration in increasing ethanol concentrations (50%, 75%, 90%, and 100%; I, II, and III; 5 min each), the slide were rendered transparent with xylene and covered with gum according to the kit instructions. Finally, the sections were imaged using a confocal scanning laser microscope (Olympus VS120).

#### 4.5.6. Transmission Electron Microscope (TEM)

The brain tissue obtained from the hippocampal CA1 region was diced into cubes (1 mm^3^), and immediately fixed in 2.5% glutaraldehyde (phosphate buffer, pH 7.2) overnight at 4 °C, and then in a mixed solution of 2.5% glutaraldehyde with 2.0% paraformaldehyde for 4 h at 4 °C. Following washing 3 times in 0.1 M PBS for 15min each, the tissues were then postfixed with 2% osmium tetroxide for 30 min and then dehydrated with a series of graded ethanol solutions. Subsequently, ethanol was substituted with propylene oxide and then embedded in Epon 812. Ultrathin sections (70 nm thick) were cut with ultramicrotome, collected on copper grids, and then stained with 4% uranyl acetate and lead citrate. Finally, the ultrastructure was observed with a transmission electron microscope (Tecnai G2 Spirit Twin), and micrographs were taken and carefully analyzed for fine structural changes. The postsynaptic density thickness was evaluated as the length of a perpendicular line traced from the postsynaptic membrane to the most convex part of the synaptic complex. The widths of the synaptic clefts (SCs) were estimated by measuring the widest and narrowest portions of the synapse and then averaging these values.

#### 4.5.7. Stereotaxic Surgery and Adeno-Associated Virus Injection

To specifically observe the dopaminergic projection of the substantia nigra to the hippocampus, we performed the same training period with DAT-cre mice instead of C57BL/6J mice. DAT-cre mice were anesthetized with 40 mg/kg sodium pentobarbital and randomly underwent stereotaxic surgery to administer the adeno-associated virus (AAV) (rAAV-hSyn-DIO-Synaptophysin-mCherry-WPRE-hGH) into the SNpc (anterior–posterior [AP]: 3 mm; medial–lateral [ML]: 1.3 mm; dorsal–ventral [DV]: 4.7 mm). A trace syringe connected to a microinjection pump was inserted into the target brain region for the viral infusion. After maintaining it in place for 2 min for stabilization, we performed infusion at 0.1 μL/minute, followed by a 15-min resting period to permit AAV diffusion. The adeno-associated virus was unilaterally injected into the SNpc. After infusion, the syringe was slowly removed, and the scalp was sutured. Mice were kept warm and provided with peanuts as a nutrient supply postoperation. After 3 weeks, we performed the perfusion and histology procedures.

### 4.6. Statistical Analysis

The statistical analysis was performed using SPSS version 25.0. All data were tested for normality and are presented as mean ± SEM. The normality of the data was assessed with the Shapiro–Wilk test. Student’s *t*-test was used to compare two groups, whereas a one-way analysis of variance (ANOVA) followed by an LSD test was used when comparing the differences between more than two groups. Linear regression was used to analyze the correlation between FNDC5 levels and the cognition performance index. A *p*-value of <0.05 was considered statistically significant.

## Figures and Tables

**Figure 1 ijms-24-07830-f001:**
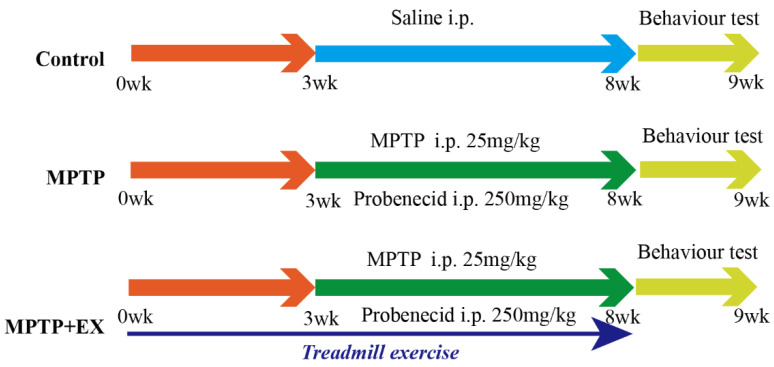
Experimental design. Control group: mice received intraperitoneal saline injection from week 3 to 8, and behavioral tests were performed at week 9. MPTP group: mice were intraperitoneally injected with MPTP (25 mg/kg) and probenecid (250 mg/kg) from week 3 to 8 (twice per week for five weeks), and behavioral tests were performed week 9. MPTP+EX: mice were subjected to treadmill exercise from week 0 to 8, MPTP was administered intraperitoneally (25 mg/kg) and probenecid (250 mg/kg) from week 3 to 8 (twice per week for five weeks), and behavioral tests were performed week 9. Some of the MPTP+EX mice were administered CTX-B or CPZ with a double 90 min interval i.p. injection three hours before euthanized. Some of the MPTP+EX mice were administered 250 μg/kg FNDC5 via i.v. injection for seven consecutive days before euthanasia.

**Figure 2 ijms-24-07830-f002:**
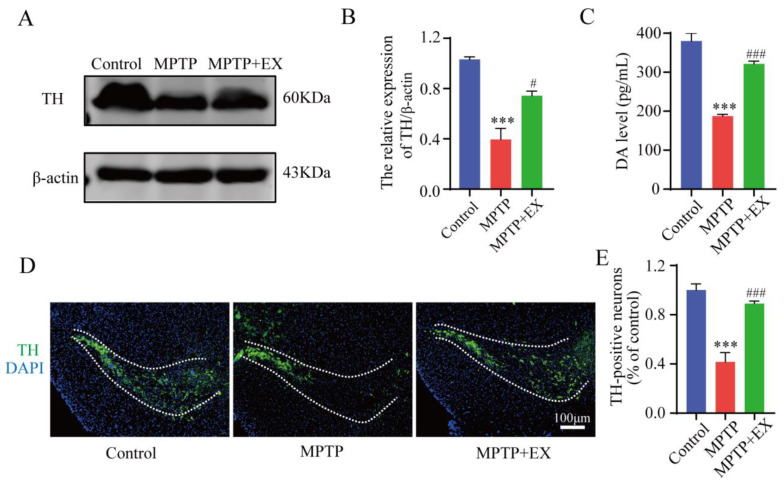
Effect of treadmill exercise on dopaminergic neuron degeneration of SNpc in MPTP-induced chronic PD model. (**A**) TH protein levels in the SNpc were examined using Western blotting (**B**) and quantification of TH levels (F(2,9) = 27.5; *p* < 0.001; n = 4). (**C**) The DA levels (F(2,33) = 1463; *p* < 0.001; *n* = 12) of the striatum were detected with ELISA. (**D**) Immunofluorescence for TH-positive neurons in the SNpc, scale bar = 100 µm, (**E**) and quantification of TH-positive neurons (F(2,15) = 120, *p* < 0.001, n = 6) in the S.N. Data represent the mean ± SEM: *** *p* < 0.001 vs. control; ^###^
*p* < 0.001 and ^#^
*p* < 0.05 vs. MPTP group.

**Figure 3 ijms-24-07830-f003:**
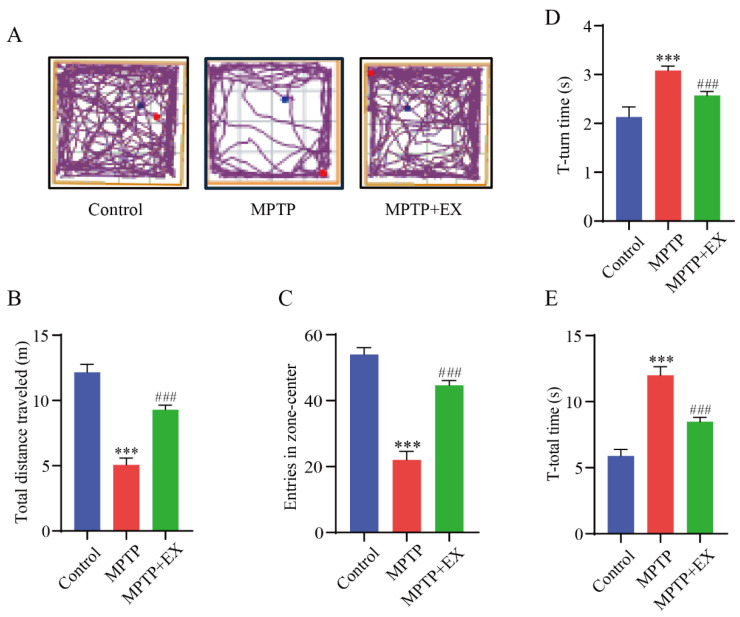
Effect of treadmill exercise on motor function in MPTP-induced chronic PD model. (**A**) An example ethogram of single-mouse locomotion features and trajectories. The blue dot is the starting position and the red dot is the ending position. (**B**) Statistics of total distance traveled (F(2,33) = 144.2, *p* < 0.001), (**C**) and entries in zone center (F(2,33) = 145.1, *p* < 0.001) of mice in the open-field test (n = 12). (**D**) Comparison of times required for T-turn (F(2,21) = 95.33, *p* < 0.001), (**E**) and T-total (F(2,21) = 290.3, *p* < 0.001) of mice in the pole test (n = 8). Data represent the mean ± SEM; *** *p* < 0.001 vs. control, ^###^
*p* < 0.001 vs. MPTP group.

**Figure 4 ijms-24-07830-f004:**
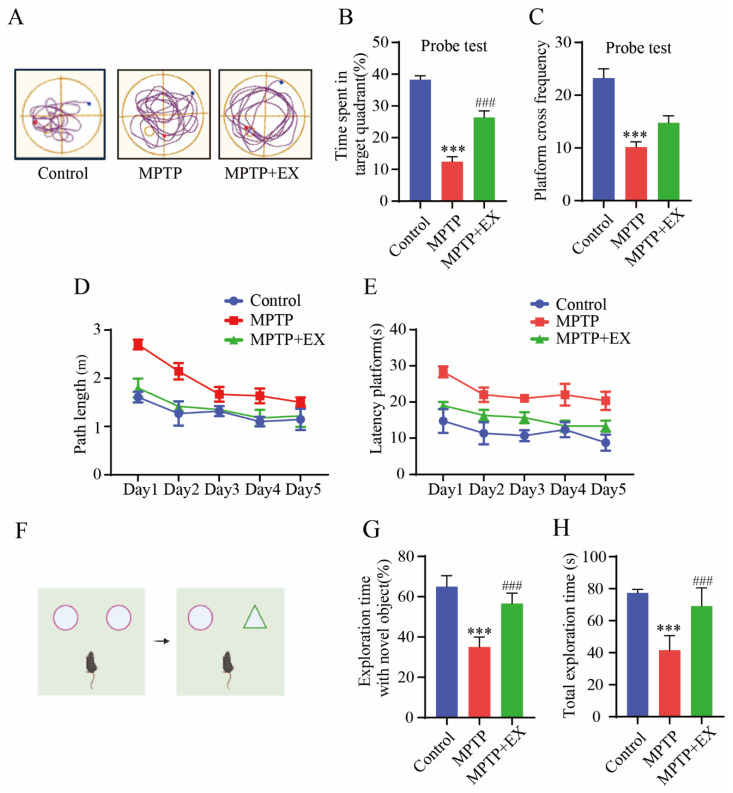
Effects of treadmill exercise on cognition dysfunction in MPTP-induced chronic PD model. (**A**) An example ethogram of single-mouse locomotion features and trajectories. The blue dot is the starting position and the red dot is the ending position. (**B**) Time spent in target quadrant (%) (F(2,33) = 58.91, *p* < 0.001), (**C**) platform-cross frequency (F(2,33) = 22.66, *p* < 0.001) (**D**) path length, (**E**) and time spent searching for platform over 5 days with respect to mice performing the Morris water maze test (*n* = 12). (**F**) Schematic diagram of novel object recognition test, (**G**) percentage of time spent with the novel object to total object exploration time (F(2,21) = 70.40, *p* < 0.001), (**H**) and the total object exploration time (F(2,21) = 39.01, *p* < 0.001) of mice in NOR (n = 8). Data represent the mean ± SEM *** *p* < 0.001 vs. control, ^###^
*p* < 0.001 vs. MPTP group.

**Figure 5 ijms-24-07830-f005:**
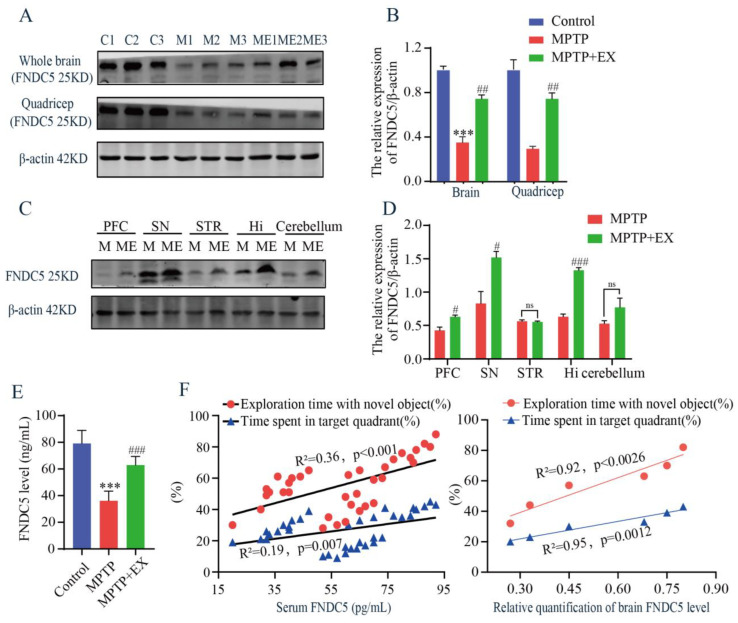
Effects of treadmill exercise on levels of FNDC5 in MPTP-induced chronic PD model. (**A**) FNDC5 protein levels in the brain (F(2,6) = 66.37, *p* < 0.001, n = 3) and quadriceps (F(2,6) = 35.15, *p* < 0.001, n = 3), (**B**) and quantification of FNDC5 levels. (**C**) FNDC5 protein levels in the PFC (t(4) = 3.902, *p* = 0.0175), SNpc (t(4) = 3.67, *p* = 0.022), STR (t(4) = 0.4936, *p* = 0.6475), Hi (t(4) = 15.03, *p* < 0.001), and CERE (t(4) = 2.651, *p* = 0.0569), n = 3, (**D**) quantification of FNDC5 levels. ns, nonsignificant. (**E**) Serum FNDC5 levels (F(2,33) = 87.50, *p* < 0.001, n = 12) were examined with ELISA. (**F**) Correlation analysis was used to analyze the association between serum-FNDC5/brain-FNDC5 levels and cognition indices (exploration time with the novel object (%) and time spent in the target quadrant (%)). Data represent the mean ± SEM; *** *p* < 0.001 vs. control, ^#^ *p* < 0.05, ^##^ *p* < 0.01, ^###^
*p* < 0.001 vs. MPTP group. C (control), M (MPTP), ME (MPTP + exercise), 1, 2, and 3 (samples from three different mice in that group), PFC (prefrontal cortex), SNpc (substantia nigra), STR (striatum), Hi (hippocampus), and CERE (cerebellum).

**Figure 6 ijms-24-07830-f006:**
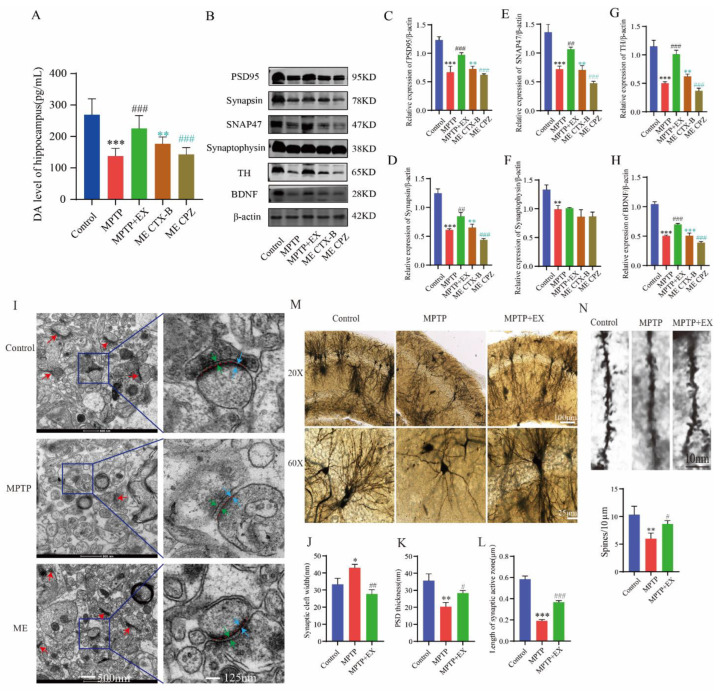
Effect of treadmill exercise on synaptic plasticity of hippocampal pyramidal neurons and dopamine levels and BDNF in MPTP-induced chronic PD mice model. (**A**) Total dopamine levels of the hippocampus (F(4,55) = 33.12, *p* < 0.001, n = 12). (**B**) Hippocampal protein levels of PSD95, synapsin, synaptophysin, SNAP47, TH, and BDNF were examined by Western blotting. (**C**–**H**) Quantification of PSD95 (F(4,10) = 53.50, *p* < 0.001), synapsin (F(4,10) = 99.18, *p* < 0.001), SNAP47 (F(4,10) = 60.12, *p* < 0.001), synaptophysin (F(4,10) = 17.47, *p* < 0.001), TH (F(4,10) = 83.36, *p* < 0.001),and BDNF (F(4,10) = 215.7, *p* < 0.001) levels. (n = 3). (**I**) Ultrastructure of synapses determined via electron micrography in hippocampus CA1 region of mice (scale bar = 500 nm). Enlarged images of the left images (scale bar = 125 nm). Red arrows indicate postsynaptic densities (PSDs), green arrows indicate the width of the synaptic cleft, blue arrows indicate the thickness of the PSD, and red dotted lines indicate the length of active synaptic zone. (**J**–**L**) Quantification of the length of the active synaptic zone (F(2,6) = 264.3, *p* < 0.001), synaptic cleft width (F(2,6) = 23.87, *p* = 0.001), and PSD thickness (F(2,6) = 21.17, *p* = 0.002), (n = 3). (**M**,**N**) Representative section of Golgi-Cox staining dendrites of CA1 pyramidal neurons from the stratum radiatum in mice. Spines number/10 μm (F(2,6) = 11.73, *p* = 0.009). Scale bars = 10 µm, 25 µm, and 100 µm; n = 3. Data represent the mean ± SEM; * *p* < 0.05, ** *p* < 0.01, *** *p* < 0.001 vs. control; ^#^ *p* < 0.05, ^##^ *p* < 0.01, ^###^
*p* < 0.001 vs. MPTP; ** *p* < 0.01, ***
*p* < 0.001 MECTX-B vs. ME; ^###^
*p* < 0.001 MECPZ vs. ME group. EX (exercise), ME (MPTP + exercise), CPZ (chlorpromazine, DA blocker), and CTX-B (cyclotraxin-B, BDNF inhibitor).

**Figure 7 ijms-24-07830-f007:**
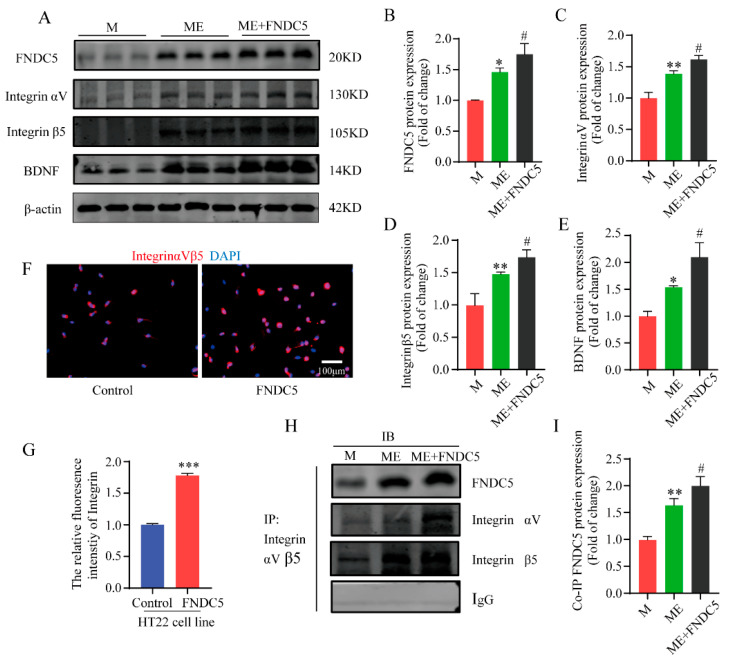
Effect of treadmill exercise on FNDC5/integrinαVβ5-mediated BDNF levels in MPTP-induced chronic PD model and verification of FNDC5–integrin αV β5 interactions. (**A**) Protein levels of FNDC5, integrin αV, integrin β5 and BDNF were examined with Western blotting. (**B**–**E**) Quantification of FNDC5 (F(2,6) = 13.59, *p* = 0.006), integrin αV (F(2,6) = 58.63, *p* < 0.001), integrin β5 (F(2,6) = 27.36, *p* = 0.001), and BDNF (F(2,6) = 33.06, *p* < 0.001) levels (n = 3). (**F**,**G**) Representative immunofluorescence images and summarized data of staining for integrin (t(10) = 33.18) in HT22 (hippocampal neuron cell line) cells. Scale bar = 100 μm, n = 6. (**H**,**I**) Interactions between FNDC5 and integrin proteins (F(2,6) = 45.77, *p* < 0.001) were examined by Co-IP, n = 3. Data represent the mean ± SEM; * *p* < 0.05, ** *p* < 0.01 *** *p* < 0.001 vs. control, ^#^ *p* < 0.05 vs. MPTP group. M (MPTP), ME (MPTP + exercise), IP (immunoprecipitation), and IB (immunoblotting).

**Figure 8 ijms-24-07830-f008:**
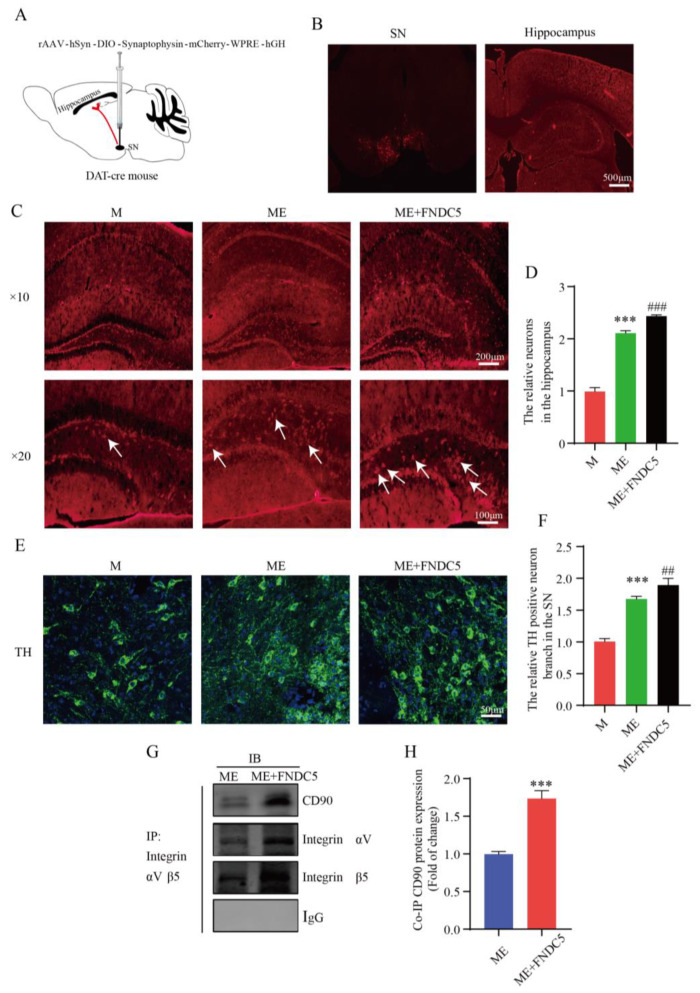
Effect of treadmill-exercise-induced FNDC5 levels on dopaminergic synaptic connections from substantia nigra to hippocampus. (**A**,**B**) Schematic diagram of virus injection and expression in SNpc and hippocampus. Scale bar = 500 μm (**C**,**D**) Representative DA neuron projection (F(2,15) = 895.2, *p* < 0.001) in hippocampus DG region of mice (scale bar = 100 µm, 200 µm. N = 6). The white arrows showed anterograde tracer virus across postsynaptic positive cells. (**E**,**F**) Representative image of TH-positive neurons and branches (F(2,15) = 223.3, *p* < 0.001) in SNpc region of mice (scale bar = 50 µm, n = 6). (**G**,**H**) Interactions between CD90 and integrin proteins in hippocampus were examined by IP (t(4) = 11.67, n = 3, *p* < 0.001). Data are presented as mean ± SEM; *** *p* < 0.001 vs. control, and ^##^ *p* < 0.01, ^###^
*p* < 0.001 vs. MPTP group. M (MPTP), ME (MPTP + exercise), ME+FNDC5 (MPTP + exercise + FNDC5), IP (immunoprecipitation), and IB (immunoblotting).

**Figure 9 ijms-24-07830-f009:**
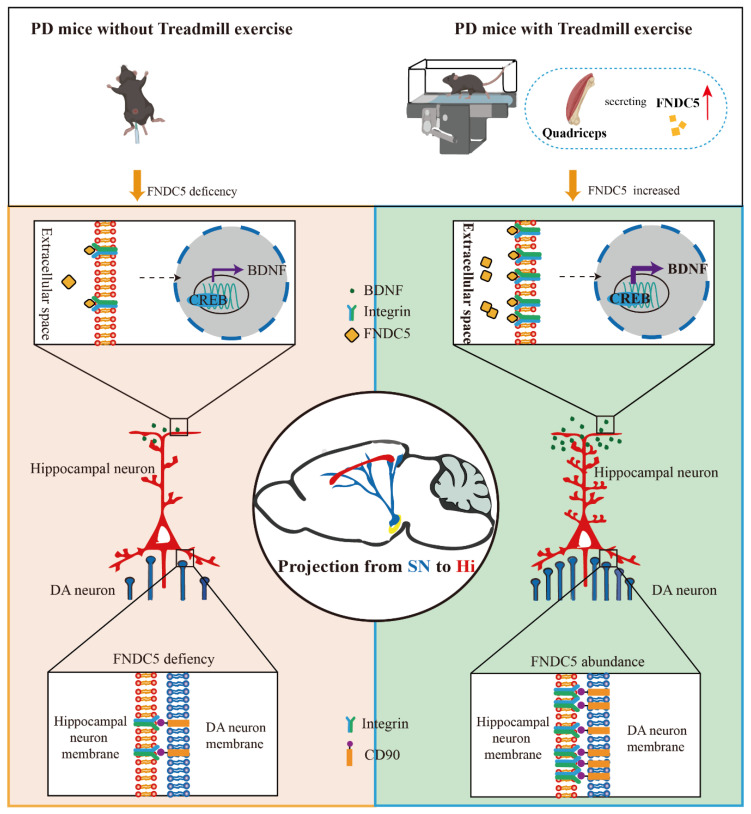
Schematic diagram of consistent exercise in PD mice promoting the synthesis of FNDC5 in the muscle and its release into the blood. FNDC5 was also u-regulated in the brain, resulting in a protective effect on the hippocampus. Direct effects: FNDC5 activated integrin αV and β5 receptors on hippocampal neurons and promoted the activation of the CREB-mediated BDNF pathway. Indirect effects: FNDC5 promoted the interaction between the integrin receptors on the hippocampal neurons and the CD90 on the dopaminergic neurons to maintain the dopaminergic synaptic connection from the substantia nigra to the hippocampus.

## Data Availability

The original contributions presented in this study are included in the article. Further inquiries can be directed to the corresponding author.

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
