# Peer review of "Treadmill Exercise Alleviates Cognition Disorder by Activating the FNDC5: Dual Role of Integrin αV/β5 in Parkinson’s Disease"

_ijms, 2023, doi:10.3390/ijms24097830_

Round 1

Reviewer 1 Report

The authors tested the hypothesis that in MPTP PD mouse model, cognitive impairment can be alleviated by treadmill exercise. They claim that FNDC5 released by exercise crosses the brain to the hippocampus, binds to the Integrin receptors and induce BDNF expression, thus the survival and neurogenesis of Substantia nigral Dopaminergic Neurons projecting to the hippocampus CA1. 

1) Overall, the study design described here is robust, though it could have been more encompassing than that described in Fig. 1.

The authors did not explain their design of the experimental group (MPTP+ EX ), in particular pre-training prior to MPTP administration. Therefore,  the evidence presented here supports treadmill exercise beneficial effects ONLY if it  preceded the onset of PD. Additional experimental groups could have been A) Exercise alone and B) MPTP full protocol followed by Exercise to test the effect of FNDC5 post PD onset. It would be helpful to expand the discussion to include this scenario.

2) There are minor linguistic errors which I eluded to some below and request that the manuscript be revised by Scientific Writing Professional for optimization. 

3)   The following references would strengthen the manuscript:

·         Ibrahim AM, Chauhan L, Bhardwaj A, Sharma A, Fayaz F, Kumar B, Alhashmi M, AlHajri N, Alam MS, Pottoo FH. Brain-Derived Neurotropic Factor in Neurodegenerative Disorders. Biomedicines. 2022 May 16;10(5):1143. doi: 10.3390/biomedicines10051143. PMID: 35625880; PMCID: PMC9138678.

 ·         Azman KF, Zakaria R. Recent Advances on the Role of Brain-Derived Neurotrophic Factor (BDNF) in Neurodegenerative Diseases. Int J Mol Sci. 2022 Jun 19;23(12):6827. doi: 10.3390/ijms23126827. PMID: 35743271; PMCID: PMC9224343.

 ·         Jiang Y, Fay JM, Poon CD, Vinod N, Zhao Y, Bullock K, Qin S, Manickam DS, Yi X, Banks WA, Kabanov AV. Nanoformulation of Brain-Derived Neurotrophic Factor with Target Receptor-Triggered-Release in the Central Nervous System. Adv Funct Mater. 2018 Feb 7;28(6):1703982. doi: 10.1002/adfm.201703982. Epub 2017 Dec 7. PMID: 29785179; PMCID: PMC5958903.

 ·         Tsetsenis T, Badyna JK, Wilson JA, Zhang X, Krizman EN, Subramaniyan M, Yang K, Thomas SA, Dani JA. Midbrain dopaminergic innervation of the hippocampus is sufficient to modulate formation of aversive memories. Proc Natl Acad Sci U S A. 2021 Oct 5;118(40):e2111069118. doi: 10.1073/pnas.2111069118. PMID: 34580198; PMCID: PMC8501778.

The Results and Methodology sections need major attention as many details are missing and shall prove confusing to the reader (please see comments below).

Line 75: Projection: it is not projection it is synaptic knobs density or contact points (sprouting)

Line 81: The results section should begin by validating the MPTP model in your lab to produce PD mouse Model. The data here is scattered throughout the figures. To put the reader at ease, it should be clearly proven to the reader that you reliably produced an MPTP PD mouse Model by including the following data: DA neurons survival histochemistry, DA levels in the Hippocampus, TH Western in the Substantia Nigra and relevant behavioral protocols. Then, you can proceed to describe your results. 

Legend Title sentence of Figures 2, 3, 4, 6, 7 and 8 need re-writing. Title sentence should describe the results NOT state conclusions. Only Figure 5 Legend title conformed to this format. Kindly edit the rest to match that of Figure 5.

The use of "MPTP PD mouse Model" need to be unified and used consistently throughout the manuscript due to the inherent limitations of this model.

Line 104 A: TH Western is overexposed and is not reliable since quantitative results are dependent on the linear part only of the band on a film. You are comparing optical densities of two bands on a film. The TH in Control appears way above saturation and therefore can not be used. you need another film of the bands that is correctly exposed and compared to B-actin.

Line 105: TH levels in which brain region?

Line 107 D: Poor DAPI image. Need to increase image size and Resolution. Also; state the objective of DAPI role.

Line 142: Re-write. Requires more relevant details so the reader is oriented to what the data mean without refrering to the Results section.

Line 383: state the rational behind using Immunoprecipitation (IP) as opposed to immunohistochemistry, which provides direct visual evidence to where the binding actually occurs? 

Line 527: Figure 9: needs a detailed legend. The top box is confusing as it conveys that only control mice receive training. I suggest you move both mice, Control and PD, to the top, followed by the treadmill below and the  Quadriceps rectangle underneath, then a bifurcate to the Box below.

Lines 564-573 (Figure 1): The authors should comment in the discussion about the ONSET time of Treadmill exercise and its possible effect. Exercise onset was BEFORE MPTP protocol. Is this the equivalent of the effect of exercise prior to PD onset? What if Exercise was used After the onset of PD pathology, will it help mitigate the effect of PD? Indeed, this is the power of the results: can exercise REVERSE PD symptoms? To address this issue the design should situate exercise AFTER MPTP protocol and not before. 

Line 564: State the purpose of Cyclotraxin-B in this design

Line 566: State the purpose of Chlorpromazine hydrochloride in this design

Line 574: why build an irisn group if the intention is to study endogenous release of irisin in vivo induced by treadmill exercise?

Line 580: State the objective of the Open Filed Test (i.e. what cognitive function does it measure)

Line 586: State the objective of the Pole Test 

Line 616: Clarify whether you mean Whole Brain tissue OR certain regions of the brain. Not clear

Line 617: why use 20 micro m brain sections? those are thick sections that include multiple layers that increases background and and could be misleading in interpretation. 

Line 629: Language issues

Line 650: Paragraph needs re-writing (not clear)

Line 652: "Brain samples were homogenized" do you mean whole brains or you mean Brain regions. If so, name regions. Then, you continue to the use the term "with" blood samples: what does that mean?

Line 655: Why measure DA in the midbrain if the hypothesis being tested here is DA projections to the Hippocampus?

Lines 357-662 Needs Re-writing

Line 663: State technique objective in this experimental design

Line 669: Define Solutions A and B

Lines 716+ 718: Language issues

Reviewer 2 Report

In this paper, Chuanxi Tang and colleagues investigated the potential beneficial effect of treadmill exercise in alleviating cognitive impairment in an MPTP mouse model of Parkinson’s’ disease via activation of the FNDC5-BDNF pathway and enhancement of dopaminergic projection to the hippocampus.

However, in the opinion of the reviewer, there is a key point that is lacking throughout the paper, and which is of paramount importance. The critical point is the definition of the level (intensity) of treadmill training (low, moderate, or high treadmill training). In fact, as reported in the current literature, distinct exercise training schedules of different intensities can cause a number of physiological adaptations in different organs, and especially at the level of the adrenal gland, which represents an early target of PE. In particular, evidence shows that the hypothalamus-pituitary-adrenocortical axis, as well as the sympatho-adrenomedullary system, is mainly involved in mediating the physiological effects of PE. Again, several morphological and biochemical changes were reported with reference to oxidative metabolism and muscle fiber composition in the mouse. In line with this, while slight to moderate levels seemed to be beneficial to cartilage health, more strenuous exercise may be detrimental. Thus, if exercise is thought to act like a drug preventing joint disease, the dosage may be critical to its success. However, from the paper, it does not clearly emerge the characteristic and intensity of treadmill PE. Do the authors have provided an incremental exercise test? If not, please comment on this point. Therefore, in the opinion of the reviewer, it is important that the Author implement the introduction by discussing and citing previous studies, some key of them reported below, and clearly discuss all these points and compare their PE treadmill protocol with previous literature to highlight whether it refers to a moderate or high-intensity treadmill training. This would add significance to the paper. Also results, data interpretation and conclusion should be revised in light of such points.

Bartalucci A, Ferrucci M, Fulceri F, Lazzeri G, Lenzi P, Toti L, Serpiello FR, La Torre A, Gesi M. High-intensity exer- cise training produces morphological and biochemical changes in adrenal gland of mice. Histol Histopathol 27, 753–769, 2012.

Toti, L.; Bartalucci, A.; Ferrucci, M.; Fulceri, F.; Lazzeri, G.; Lenzi, P.; Soldani, P.; Gobbi, P.; La Torre, A.; Gesi, M. High-intensity

exercise training induces morphological and biochemical changes in skeletal muscles. Biol. Sport 2013, 30, 301–309

Shimomura S, Inoue H, Arai Y, Nakagawa S, Fujii Y, Kishida T, Ichimaru S, Tsuchida S, Shirai T, Ikoma K, Mazda O, Kubo T. Treadmill Running Ameliorates Destruction of Articular Cartilage and Subchondral Bone, Not Only Synovitis, in a Rheumatoid Arthritis Rat Model. Int J Mol Sci. 2018 Jun 3;19(6):1653. doi: 10.3390/ijms19061653. PMID: 29865282; PMCID: PMC6032207.

Zhou X, Cao H, Wang M, Zou J, Wu W. Moderate-intensity treadmill running relieves motion-induced post-traumatic osteoarthritis mice by up-regulating the expression of lncRNA H19. Biomed Eng Online. 2021 Nov 18;20(1):111. doi: 10.1186/s12938-021-00949-6. PMID: 34794451; PMCID: PMC8600697

Another major point is that the paper needs a substantial revision of the English language and style, possibly by a native speaker.

A spell check is required throughout the manuscript.

Furthermore, there are several minor points, and some of them are reported below.

Title: Please consider changing the term “cognition disorder” to “cognitive impairment”

Abstract:

Line 16: please delete “Disruption of”

Line 16: please change “cognition” to “cognitive” (also in all other parts of the paper)

Lines 20, 21 (and so on): correct “Exercise” to “exercise”. Why is this term always reported with an initial capital letter? Is there any significance for that?

Lines 22-23: FNDC5 is the precursor of the hormone; please correct.

Line 25: mouse model

Line 32: “Further” is written in bold. Please delete the space between the two parts of the abstract.

The corrections suggested in the abstract should be reported consistently throughout the text.

Line 68: “substantia nigra (SN) region”; please consider changing this to “substantia nigra pars compacta (SNpc)

Line 69: “meaning anatomical fibers”; this is not clear to the reader; please revise.

Results:

There is no Figure 1. Please revise both the text and the figure legend.

Figure 2. Authors should implement the quality or the magnification of the representative pictures of Immunofluorescence for TH-positive neurons in the SN. The DAPI positive cells are poorly visible.

Figure 6: panel I, figure legend, please correct red, blue, green arrows (not rows)

Materials and Methods:

Overall the materials and methods are adequately described. In the first paragraph, it should be clearly reported how many animals per group (control, MPTP, MPTP+EX) were used.

Why the EX group was not included (as instead was done in Authors’ previous manuscript?)

Discussion: even this section requires a revision of English language and style.
